# Implementation of a CAM Assay Using Fibrosarcoma Spheroids

**DOI:** 10.3390/ijms26115318

**Published:** 2025-05-31

**Authors:** Flemming Puscz, Noah Jozsef Hatem, Sonja Verena Schmidt, Felix Reinkemeier, Marius Drysch, Mustafa Becerikli, Yonca Steubing, Marcus Lehnhardt, Christoph Wallner

**Affiliations:** Department of Plastic Surgery, BG University Hospital Bergmannsheil, Ruhr University Bochum, 44789 Bochum, Germany; noahhatem@outlook.de (N.J.H.);

**Keywords:** CAM assay, fibrosarcoma, soft tissue sarcoma, HT1080

## Abstract

Fibrosarcomas represent a rare but highly aggressive tumor entity among soft tissue tumors. Due to its rarity, questions regarding its development and pathophysiology remain unclear. The chorioallantoic membrane (CAM) assay represents an easily available method to investigate tumors on a growth membrane, live and in ovo. The following study was established to test whether the growth of fibrosarcoma spheroids on the CAM was possible and to critically review their applicability for downstream investigations. The shells of fertilized chicken eggs were opened and the previously prepared HT1080 cell spheroids (50,000, 75,000, and 100,000 cells per spheroid) were applied to the CAM. After 7 days, tumors were examined for size, weight, and vascularization. After 7 days, 80 of 163 chicken eggs showed sufficient tumor growth. Of these 80 eggs with confirmed tumor growth, 32 (40%) were from the 50,000 spheroid group, 18 (22.5%) were from the 75,000 spheroid group, and 30 (37.5%) were from the 100,000 spheroid group. The 100,000-cell spheroid group exhibited the highest weights, with a mean of 110.7 mg, as well as tumor size expansion. This cell number also showed the highest vascularization rates. Tumor growth of fibrosarcoma spheroids could successfully be initiated on the CAM. Consequently, the CAM assay presents a good base for future studies involving human fibrosarcoma cell spheroids.

## 1. Introduction

Fibrosarcoma is a rare and aggressive subtype of soft tissue sarcomas. Although soft tissue sarcomas themselves make up a low percentage of adult malignant tumors, fibrosarcomas account for just 3.6% of all sarcomas [1]. It is most common in middle-aged adults, presumably slightly more common in men, within the age range of 40–55 years. It typically occurs in soft tissues, specifically the tendons and fascia of the extremities, trunk, head, and neck, but can also originate from bones [2,3]. The prognosis for fibrosarcomas is complex and depends on several factors, including patient age, tumor size, penetration of surrounding structures, metastatic potential, and histopathological grading. It has been described in the literature that 80% of fibrosarcomas are classified as high-grade, whereas 25% of the remaining tumors also develop into high-grade sarcomas. This, combined with a high percentage of local recurrences and metastatic behavior, explains the survival rates of less than 70% at 2 years and less than 55% at 5 years [4]. Fibrosarcomas originate from spindle cells; therefore, they are of mesenchymal origin and exhibit uncontrolled proliferation [2]. Genetically, multiple non-specific numerical and chromatic aberrations can be detected. The standard therapy for fibrosarcomas is surgery. Depending on the tumor size, localization, and infiltration, the surgical procedure will be determined. Intramuscular localized tumors are resected en bloc, whereas larger resections with R0 margins should be achieved if the tumor has not reached the muscle origin or insertion, or grown beyond the compartment. Deep, high-grade tumors larger than 5 cm are an indication for adjuvant radiation therapy. The necessity of adjuvant radiotherapy in complex constellations of tumor size, grade, and localization should be discussed by a multidisciplinary team [5]. In this context, the role of neoadjuvant chemotherapy should also be discussed [6].

The avian chorioallantoic membrane (CAM) model is a simple, cost-effective, and time-efficient research model used to investigate tumor growth and tumor angiogenesis. During avian embryo development, the fusion of the mesodermal layers of the allantois and chorion occurs, resulting in the formation of the so-called chorioallantoic membrane. During development, the CAM undergoes a tenfold increase in surface area and develops a robust vascular network and lymphatic system. The CAM takes on the function of the embryonic lung, allowing gas exchange through the eggshell, and acts as a reservoir for the waste products of circulation, such as urea. Since the chicken embryo is immunodeficient, allo- and xenogeneic cells are not rejected. Furthermore, implanted tumor cells do not suffer cell damage, experience sufficient vascularization, and are likely supplied with growth factors [7].

The growth behaviors of HT1080 cells have previously been investigated in various experimental studies [8,9,10]. What these studies have in common is the use of cell suspensions on the CAM. To our knowledge, there are no studies describing the use of human fibrosarcoma spheroids in the CAM assay. Yet, spheroids from other cell entities have already been successfully used on the CAM [11]. Given the ongoing need for in vivo-like tumor models for fibrosarcoma research, we designed this study as a model-establishing investigation, aiming to determine whether spheroid-based implantation is technically feasible and biologically effective. Furthermore, by using the CAM assay as a “non-animal” experiment, this model represents an opportunity to avoid animal experiments in the spirit of the principles of replacement, reduction, and refinement (3Rs) [12].

A cell spheroid is a three-dimensional cell culture model that mimics the structural and functional characteristics of tissues or organs in the human body [13]. It consists of a cluster or aggregate of cells that self-assemble and form a spherical shape, resembling a small-scale version of a tissue. They are created by allowing cells to grow and interact with each other in a controlled environment. Typically, cells are seeded onto a substrate or in suspension, and over time, they aggregate and form a compact sphere. This process is facilitated by the cells’ ability to communicate and adhere to each other, leading to the formation of cell–cell contacts. One of the advantages of cell spheroids is that they better represent the physiological conditions found in the human body compared to traditional two-dimensional cell cultures [14]. In two-dimensional cultures, cells grow as a monolayer on a flat surface, which does not accurately mimic the complex cell–cell and cell–matrix interactions that occur in tissues [15]. In our understanding, cell spheroids provide a more realistic microenvironment, allowing cells to exhibit more natural behaviors, such as enhanced cell–cell signaling, nutrient and oxygen gradients, and resistance to external stresses.

In this study we analyzed the effect of different tumor spheroid cell sizes on growth behavior, regarding size, weight, and vascularization of the tumor, with the aim of optimizing the use of spheroids on the CAM assay for fibrosarcoma research.

## 2. Results

### 2.1. Egg Development and Tumor Growth

Of the 220 eggs acquired, 34 (15.5%) showed no fertilization. These were therefore discarded immediately after assessment. During CAM preparation, another 23 (10.5%) eggs were excluded from the experiment because either the CAM did not develop regularly or a CAM vessel was injured during the operation. Thus, a total of 163 (74.1%) eggs could be inoculated with spheroids (Figure 1a,b). Among them, 62 eggs were inoculated with 50,000, 38 eggs with 75,000, and 63 eggs with 100,000 cells. During daily follow-up, all fertilized eggs were carefully monitored for the development of complications, with particular attention paid to albumen leakage, contamination, and fungal infections. As part of this routine assessment, 16 of the 163 inoculated eggs (9.8%) exhibited one or more of these complications and were therefore excluded from further analysis. Consequently, 147 of the initially prepared 220 eggs (66.8%) successfully reached the end of the experimental period on day 17. During the opening of the eggs, the inoculation site on the CAM was examined macroscopically and microscopically for the presence of a tumor by an experienced sarcoma surgeon. This showed definite tumor growth in a total of 80 (49.1%) eggs. Of these 80 eggs with confirmed tumor growth, 32 (40%) were from the 50,000 spheroid group, 18 (22.5%) were from the 75,000 spheroid group, and 30 (37.5%) were from the 100,000 spheroid group (Figure 1c). Based on the respective spheroids, 51.6% of the 50,000 spheroid group showed tumor growth. In the 75,000 spheroid group, 47.4% showed tumor growth, and in the 100,000 spheroid group, 47.6% showed tumor growth.

### 2.2. Tumor Area

Tumor area was expressed in cm^2^ using the two longest tumor dimensions (Figure 2). This showed that the group of 100,000-cell spheroids formed the largest area, with a mean of 0.43 cm^2^ (95% CI: 0.3 to 0.55 cm^2^). These were significantly larger than those of the 50,000 (mean: 0.22 cm^2^, 95% CI: 0.16 to 0.29 cm^2^) and 75,000-cell spheroids (mean: 0.15 cm^2^, 95% CI: 0.08 to 0.21 cm^2^).

### 2.3. Tumor Weight

In terms of tumor weight (Figure 3), the 100,000 spheroid group exhibited a mean weight of 110.7 mg (95% CI: 59.64 to 161.7 mg), which was significantly higher than the mean weight of the 75,000 spheroid group (mean: 34.34 mg, 95% CI: 21.64 to 47.04 mg). The mean weight of the 50,000 spheroid group was 49.88 mg (95% CI: 31.81 to 67.95 mg).

### 2.4. Quantification of Vascularization

In terms of efferent and afferent vascular connections to vascularize the growing tumor, the largest spheroids were found to exhibit the most vascular connections (Figure 4). Among them, the 100,000-cell spheroids exhibited an average of 9.25 (95% CI: 7.87 to 10.63) vascular connections, the 75,000-cell spheroids exhibited an average of 7 (95% CI: 5.83 to 8.17), and the 50,000-cell spheroids exhibited an average of 5.3 (95% CI: 4.72 to 5.89).

### 2.5. Immunohistochemistry and Immunofluorescence

The prepared hematoxylin and eosin, vimentin, and Ki67 sections were submitted to an experienced clinical pathologist for review. The HE sections (Figure 5a) served as a general overview. The vimentin sections showed clear positivity of the tumor tissue for the antibody (Figure 5b), as also found in human clinical samples. For the immunofluorescence sections stained with Ki67, there was also clear positivity of the tissue for the antibody (Figure 6). Thus, in the samples of the three groups, all tumor samples were unequivocally identified as fibrosarcomas.

## 3. Discussion

The aim of this study was to critically evaluate the feasibility of growing solid fibrosarcomas on the CAM with the help of cell spheroids, which was what we were able to demonstrate. The cultivation of HT1080 fibrosarcoma cells applied on the CAM in a cell suspension has already been demonstrated in preliminary works [8,9,16]. For example, Shimo et al. investigated the influence of connective tissue growth factor (CTGF) on the neovascularization of solid fibrosarcomas [16]. They grew solid tumors from 5 × 10^6^ HT1080 cells in immunodeficient Balb/c nude mice. The tumors were then removed and applied to the CAM to study neovascularization. They demonstrated that when CTGF was inhibited by IgG antibodies, neovascularization was strongly inhibited. The method used to study angiogenesis here was a qualitative, double-blinded assessment of the density of capillary vessels under a light microscope [16]. In another study, Deryugina and colleagues investigated metastasis, as well as intravasation, of different HT1080 variants [17]. The isogenic cell lines that were used disseminated either strongly or weakly. They found that the strong-disseminating HT1080 cells were able to leave the primary tumor site and spread to the periphery via the vasculature, i.e., they metastasized. The weakly disseminating cells, on the other hand, could not leave the primary tumor and remained in place. Interestingly, this effect was bypassed when the cells were applied directly into the vessels. In addition, they examined the effect of various matrix metalloproteinases (MMPs) on intravasation and demonstrated that the down-regulation of MMP-9 elicited a threefold increase in intravasation ability in the strong-disseminating HT1080 cells [17]. In further studies, this research group was also able to demonstrate the reversibility of this MMP-9 or neutrophil-triggered angiogenesis by anti-inflammatory agents (ibuprofen and cortisone) [18]. Inflammatory leukocytes, namely neutrophil-like heterophils, played a supporting role in tumor angiogenesis. Furthermore, in the CAM model, the tumor-promoting effect of ethanol on HT1080 cells was demonstrated. This effect was detected at the mRNA level by increased detection of vascular endothelial growth factor (VEGF), as well as in the final tumor volume [19]. Thus, as can be seen from the literature review, some well-established models using HT1080 cells already exist. What distinguishes our work is the systematic development and testing of fibrosarcoma-specific 3D spheroids in the CAM assay—a combination not yet described in the literature. We were able to demonstrate that the resulting tumors were morphologically recognizable as such and exhibited active vascularization. 

As described in previous papers [20,21], CAM experiments usually require relatively high numbers of cases with more than 100 eggs. In the present work, 220 eggs were used initially, of which 80 were included in the final evaluation of the experiment. The reasons for this relatively low final yield are manifold: on the one hand, not all eggs were fertilized, which is why these unfertilized eggs did not form the CAM. Again, our fertilization rates were consistent with those described previously [19]. Furthermore, after opening the eggshell, there is a risk of infection, especially by fungi. In our experiments, the eggs were cleaned of superficial contamination with Millipore water and disinfected with mucosa-friendly antiseptic before being placed in the incubator, but this can still only produce a low-germ environment. In our preliminary tests, various disinfectants were also used in this regard, but all exhibited a similar failure rate due to fungal infections. One approach to reduce the infection rate is to use sterile eggs, but these have significantly higher costs (approx. 0.50 € for a standard egg vs. approx. 3.00 € for a sterile egg). However, this would significantly increase the cost of trials with such high numbers of cases, thereby undermining one of the main arguments for this model—its cost-effectiveness. Nevertheless, contamination is an uncommon complication in this model. Fungal infections occurred in our case, especially during the first series of experiments. This problem was no longer present in the last series of experiments. Yet, this model represents a way to reduce in vivo animal testing.

In terms of cell number, the 100,000-cell spheroids exhibited an advantage across all study points. Although the smaller cell numbers also showed adequate tumor growth, the tumors from the 100,000-cell spheroids were significantly larger and heavier, which also facilitated a clinically unambiguous identification of the tumor. An even higher cell number does not seem to be target-oriented due to the expected hypoxia and necrosis in the core. Regarding the results on tumor size and weight in the 75,000-cell group, we expected them to fall between those of the 50,000- and 100,000-cell groups. Why this group delivered the smallest and lightest results remains unclear to us and may be evaluated in further work. We deliberately chose to apply cell spheroids because they are three-dimensional constructs, and they have a closer relationship to in vivo tumor development. The application of HT1080 cell suspensions would be one way to apply even more tumor cells to the CAM. However, this could complicate the identification of the tumors in ovo. This experimental aspect should be addressed in the future as it has already been explored in retinoblastoma research [22].

Regarding the efferent and afferent vessels of the tumor, we were able to demonstrate that the inoculated fibrosarcoma spheroids trigger sufficient new vessel formation and enable tumor growth. Again, the 100,000-cell spheroids showed the greatest number of vascular connections, which makes sense when viewed together with tumor area and weight. Regarding the quantification of this vascularization, different investigation approaches exist [12]. Among others, immunohistochemical staining with lectin [23], ink injection [24], or, as in our case, the counting of vascular connections are described. This method has been explained before [25], and there are also (semi-)automatic algorithms for this procedure [26,27]. Manual counting, as we used, may lead to errors in distinguishing between true blood vessels and artifacts, despite blinding of the examiner to spheroid size. Future studies could benefit from the integration of automated image analysis or other quantitative methods to enable more precise and reproducible assessments of vascularization in the CAM assay. Nevertheless, we initially chose this quantification method because it is the most readily available and cost- and time-efficient method.

We are aware that histological processing of the specimens was performed on only a very small number of tumors. Therefore, no further data analysis was performed here. However, what we wanted to show was the identification of the grown tumors as fibrosarcomas on a histological level. Of course, in further experiments, histological processing of more tumor samples should be performed. In addition, these cultured tumors could then be compared to human fibrosarcoma samples.

HT1080 fibrosarcoma cells are a commonly used and universally available immortalized cell line that has been a driver of experimental sarcoma research for several decades. These cells are regularly used both in vitro [28,29] and in vivo [30,31,32] to study new therapeutic approaches, as well as tumor development. Here, the CAM model represents a possible link between in vitro and in vivo experiments. The cells can be pre-treated with a wide variety of therapeutic agents (chemotherapeutic agents, radiotherapy, etc.) prior to inoculation onto the CAM, and subsequently their growth can be evaluated in a short, but as we have shown, sufficient experimental period. Subsequently, the resulting tumors can be further examined clinically, histologically, and molecularly. In conclusion, this study represents a foundational step toward establishing a reproducible, cost-effective, and ethically favorable CAM-based platform for fibrosarcoma research. Although the findings may appear preliminary in nature, our model offers a flexible basis for future applications, such as pharmacological testing, angiogenesis modulation, or molecular profiling. Follow-up experiments utilizing this system are necessary, with the long-term goal of refining early-phase sarcoma research while contributing to the reduction of animal experimentation.

## 4. Materials and Methods

### 4.1. Experimental Groups and Study Design

Avian eggs were divided equally after initial incubation for 10 days and were inoculated with three different spheroid sizes (50,000, 75,000, and 100,000 cells). During the 7-day tumor growth period, the eggs were monitored daily for possible complications (leaky eggshells, fungal infections, etc.) and were removed from the experiment if necessary. On day 17, photo documentation of the tumors, sampling, and measurements were performed.

### 4.2. Cell Line and Spheroid Formation

HT1080 human fibrosarcoma cells were purchased from the American Type Culture Collection (ATCC, Wesel, Germany). This established cell line has already been used in preliminary experiments [33,34]. In brief, the cells were thawed and conditioned for 7 days in DMEM containing 1% L-glutamine (Merck KGaA, Darmstadt, Germany) and 10% fetal bovine serum (Pan-Biotech GmbH, Aidenbach, Germany). The cells were grown to a sub-confluent monolayer and maintained at 37 °C in a humidified atmosphere. During this phase, the cells were monitored daily using light microscopy. The medium was changed depending on the cell number, adherence, and quality of the cell medium. After sterilization, 150 µL of warmed agarose was added to each well of a 96-well plate. After cooling of the agarose layer and based on the intended cell number per spheroid (50,000, 75,000, or 100,000 cells), the corresponding cell suspension was prepared and seeded onto the agarose-coated 96-well plate to induce spheroid formation (Figure 1b). The formed spheroids were inoculated onto the CAM on day 3–5 after seeding.

### 4.3. Chorioallantoic Membrane Assay

CAM preparation was performed according to previous protocols (Figure 7) [35,36]. A total of 220 fertilized chicken eggs from a commercial chicken farm (Hof Brinkschulte GmbH & Co. KG poultry farm, Senden, Germany) were purchased for the experiments. Before being placed in the incubator, the eggs were cleaned of coarse impurities using Millipore water and were disinfected with a mucosa-friendly antiseptic (Octenisept, Schülke & Mayr GmbH, Norderstedt, Germany). These were initially incubated for 10 days at 37 °C and 50% humidity, with constant, slow rotation. On day 10, the eggs were candled for the presence of a chicken embryo. In the presence of the embryo, the most prominent blood vessels were marked, and the egg was included in the further experiments. For this purpose, the eggshell was fenestrated around the previously marked blood vessels with a straight grinder on an area of about 5 cm^2^ to dissect the CAM (Figure 1a). If blood vessels were injured in the process, the egg was removed from the experiment. The CAM was then inoculated with the prepared fibrosarcoma spheroids. The fenestration was sealed with plaster tape and the eggs were incubated for another 7 days at 37 °C and 50% humidity without rotation. Since the eggs used were non-sterile eggs and the preparation environment was low in germs but not germ-free, the eggs were monitored daily for the presence of fungal infections or other complications. If a complication was present, the affected eggs were removed from the experiment. On day 17, the eggs were inspected for possible tumor growth. For this purpose, they were cut into halves, the embryo was sacrificed, and the grown tumors were photodocumented in ovo (Figure 1c). The tumors were then dissected out, measured, weighed, and stored in 4% paraformaldehyde for further histological processing.

### 4.4. Weight and Tumor Surface Measurements

After dissection, the tumors were measured using light microscopy to calculate the tumor area. Subsequently, they were weighed three times on a precision balance (Sartorius Cubis, Sartorius, Goettingen, Germany), and the mean value was calculated.

### 4.5. Vascularization

To determine vascularization, the grown tumors were photodocumented in ovo. Images were obtained in a standardized setting with equal distances and at 5× magnification (ZEISS Stemi 305, Carl Zeiss, Oberkochen, Germany). Subsequently, the images were analyzed using ImageJ Software (Version 1.54g, National Institute of Health, Bethesda, MD, USA). The efferent and afferent vascular connections of the tumor were counted (Figure 8). Only vessels with a minimum diameter of 0.5 mm were counted. All observations were made blinded of the spheroid cell number.

### 4.6. Histology, Immunohistochemistry, and Immunofluorescence

To demonstrate that the resulting tumors on the CAM were fibrosarcomas, two representative tumors were harvested from each of the three groups, temporarily stored in 4% paraformaldehyde, and embedded in paraffin for 48 hours. Thereafter, the specimens were sectioned using a microtome at 8 μm. To obtain a general overview of the specimens, they were stained using hematoxylin and eosin. To detect fibrosarcoma cells, we used immunohistochemical vimentin staining (Thermo Fisher Scientific Inc., Waltham, MA, USA) at a dilution of 1:200. Immunofluorescence staining using Ki67 antibodies (Acris Antibodies GmbH, Herford, Germany) was also carried out on 8 μm tumor sections. For immunofluorescence, blocking was performed using 10% normal goat serum (AbCam, Cambridge, United Kingdom) in phosphate-buffered saline for one hour. The slides were stained with primary antibodies (dilution 1:200). After overnight incubation, secondary antibody Alexa Fluor 488 (Thermo Fisher Scientific Inc., Waltham, MA, USA) was applied and the slides were incubated for 1 h. The sections were counterstained using DAPI (Thermo Fisher Scientific Inc., Waltham, MA, USA) and were mounted with fluorescence mounting medium. All antibodies were used according to the manufacturer’s instructions and previous protocols [37]. The sections were then examined by an experienced sarcoma pathologist and assessed for the presence of fibrosarcoma cells. The light microscopic and immunofluorescence images were taken using the Keyence BZ-X810 (Keyence DEUTSCHLAND GmbH, Neu-Isenburg, Germany) at 4× and 20× magnification.

### 4.7. Statistical Analysis

Data were collected using Microsoft Excel (Version 16.73, Microsoft, Redmond, WA, USA). Statistical analysis was carried out using GraphPad Prism 9.5.1 (GraphPad Software, La Jolla, CA, USA). The presence of a normal distribution was tested using the Shapiro–Wilk test. Since no normal distribution was found, further analysis was carried out using one-way analysis of variance (Kruskal–Wallis–ANOVA) and Dunn’s post-hoc test. All values were expressed as means, with 95% confidence intervals (95% CI) in brackets. A *p*-value of <0.05 was considered significant.

## 5. Conclusions

This study demonstrates the successful application of the CAM assay for cultivating fibrosarcoma spheroids, highlighting its potential as a cost-effective and ethically advantageous model for preclinical research. Among the tested spheroid sizes, those with 100,000 cells exhibited the most robust tumor formation, with superior size, weight, and vascularization. Although technical limitations remain, such as handling precision and contamination, the model provides valuable insights into fibrosarcoma growth and vascular dynamics. Future optimization could strengthen its role as a translational platform bridging in vitro and in vivo sarcoma research, supporting the refinement and reduction of animal experimentation.

## Figures and Tables

**Figure 1 ijms-26-05318-f001:**
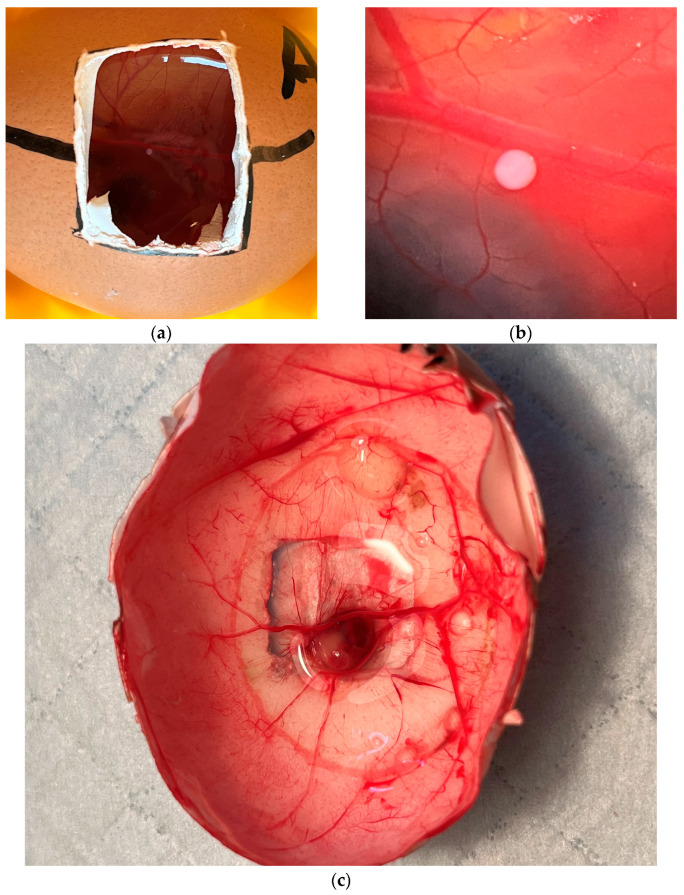
(**a**) Macroscopic image of the inoculated 100,000 HT1080 cell spheroid near a CAM vessel; (**b**) light microscopic of the same spheroid; and (**c**) fibrosarcoma growth on day 17 in ovo.

**Figure 2 ijms-26-05318-f002:**
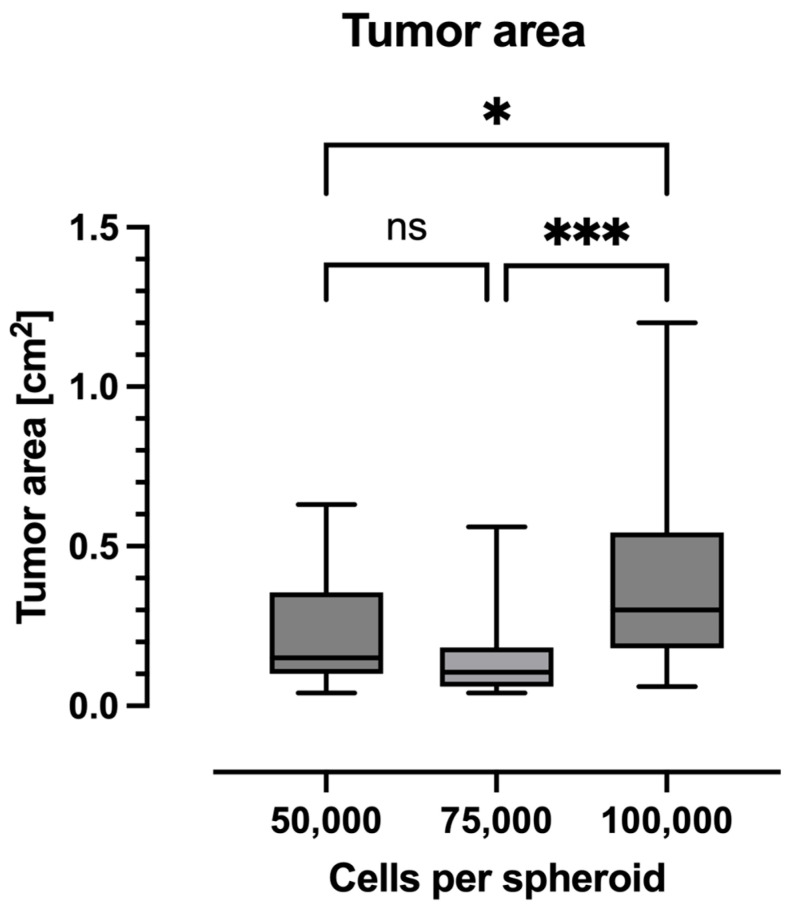
Data are presented as boxplots. The boxplots show the media and the 25% and 75% quartile; the whiskers indicate the minimum and maximum values. The tumors grown from 100,000-cell spheroids demonstrated a significantly higher tumor area in comparison to the 50,000 and 75,000-cell spheroids (*: *p* < 0.05; ***: *p* < 0.001; ns: no significance).

**Figure 3 ijms-26-05318-f003:**
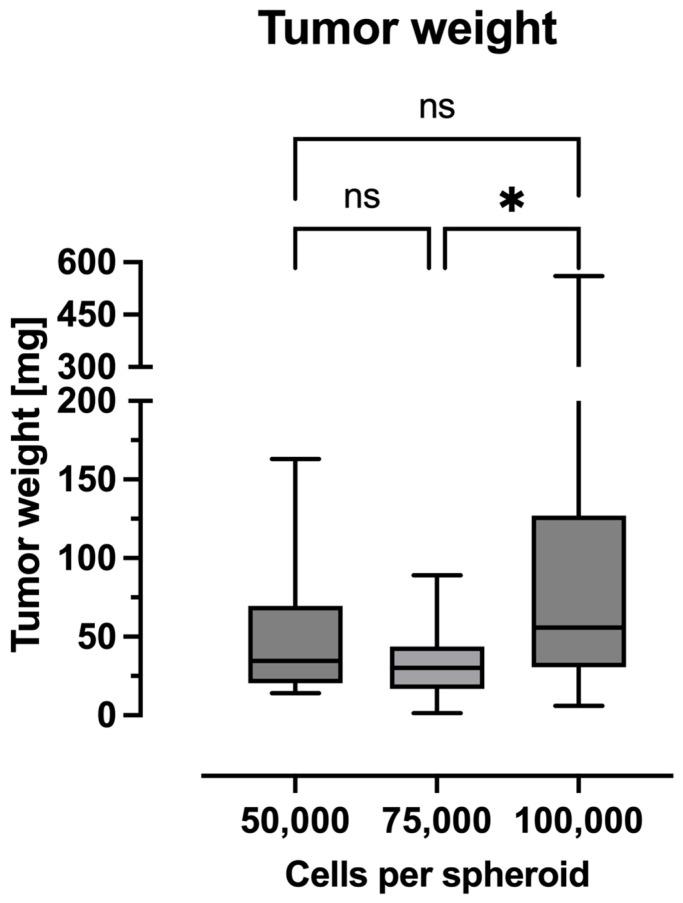
Data are presented as boxplots. The boxplots show the median and the 25% and 75% quartile; the whiskers indicate the minimum and maximum values. The tumors grown from 100,000-cell spheroids demonstrated a significantly higher tumor weight in comparison to 75,000-cell spheroids (*: *p* < 0.05; ns: no significance).

**Figure 4 ijms-26-05318-f004:**
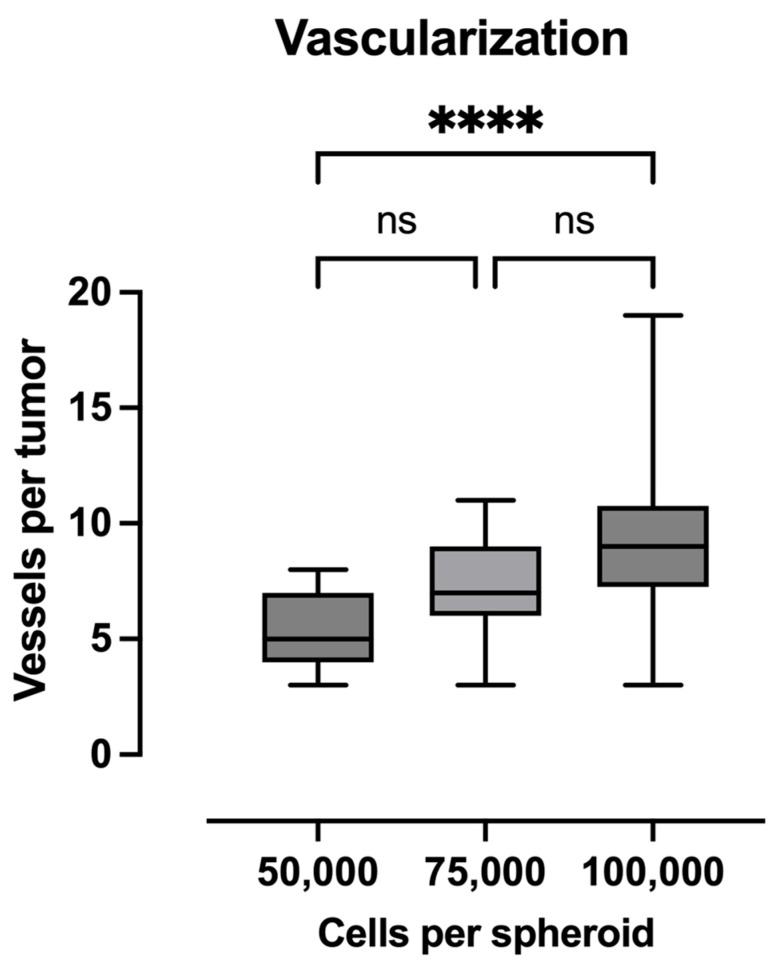
Data are presented as boxplots. The boxplots show the median and the 25% and 75% quartiles; the whiskers indicate the minimum and maximum values. The tumors grown from 100,000-cell spheroids demonstrated significantly higher vascularization in comparison to 50,000-cell spheroids (****: *p* < 0.0001; ns: no significance).

**Figure 5 ijms-26-05318-f005:**
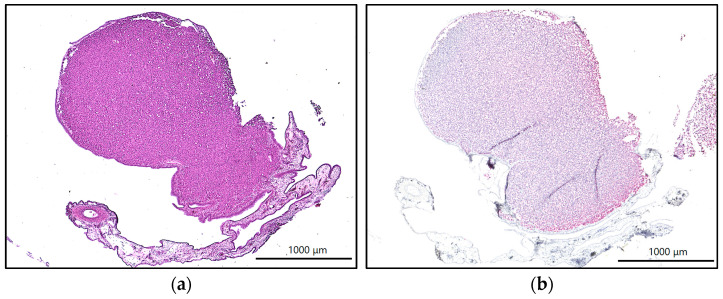
(**a**) Hematoxylin and eosin section of a fibrosarcoma grown from a 100,000-cell spheroid; (**b**) vimentin staining of the same preparation, with positivity for the antibody.

**Figure 6 ijms-26-05318-f006:**
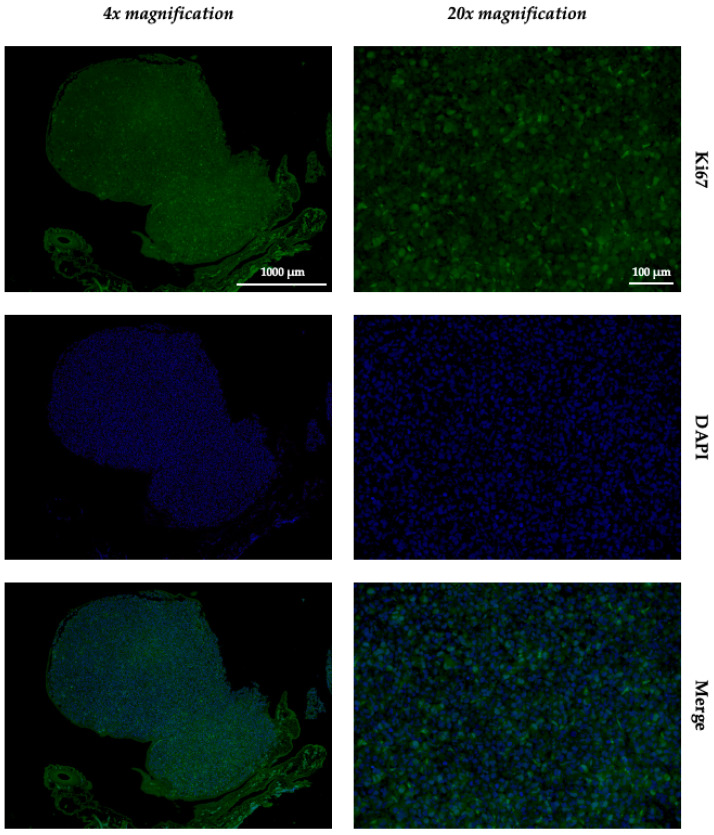
Immunofluorescence images of Ki67 staining of the same fibrosarcoma grown from a 100,000-cell spheroid at 4× (**left**) and 20× (**right**) magnification.

**Figure 7 ijms-26-05318-f007:**
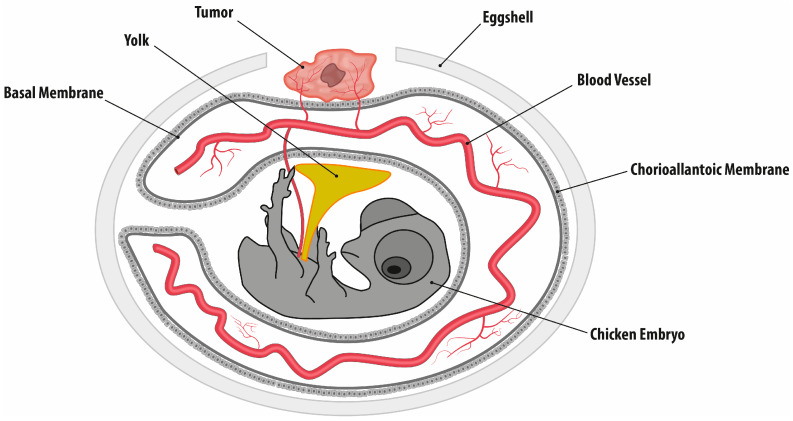
CAM model setup. Shown is the experimental setup on day 17. The opened eggshell allows access to the CAM, on which a vascularized fibrosarcoma grows.

**Figure 8 ijms-26-05318-f008:**
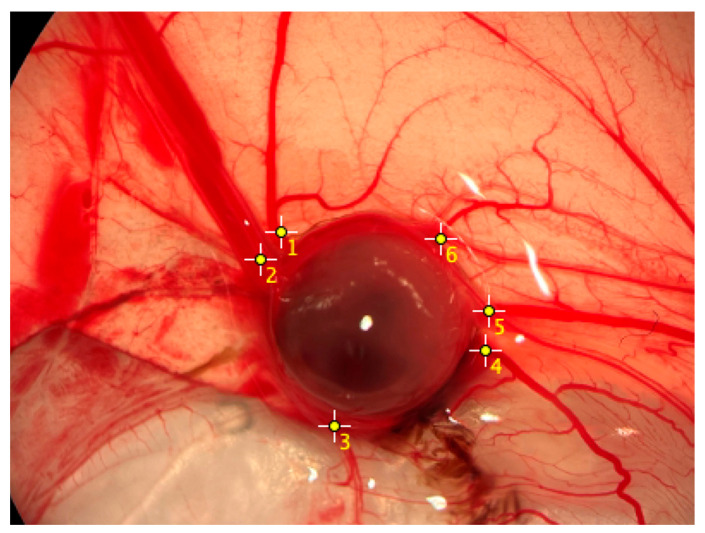
Representative image of vascular connections in the CAM. Macroscopic visualization of a fibrosarcoma tumor formed from a 100,000-cell spheroid, showing afferent and efferent vascular connections on the CAM (day 17). The yellow crosses indicate vessels connected to the tumor mass.

## Data Availability

The raw data supporting the conclusions of this article will be made available by the authors on request.

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
