# Peer review of "Implementation of a CAM Assay Using Fibrosarcoma Spheroids"

_ijms, 2025, doi:10.3390/ijms26115318_

Round 1
Reviewer 1 Report
Comments and Suggestions for Authors
In the manuscript “Implementation of a CAM assay with usage of Fibrosarcoma Spheroids” the authors show that they have successfully established the CAM model with spheroids of the HT1080 cell line for the evaluation of fibrosarcomas. The study is well conducted, the results are interesting, and the overall presentation of the data is satisfactory. However, some concerns have to be addressed.
In my eyes, the main problem of the the study is that it seems to report preliminary results, like a pilot study. HT1080 cells have been used in a CAM model before, just not as spheroids, and spheroids of tumor cells have been used before in a CAM model, just not from fibrosarcomas. Don’t get me wrong, this is important work and I congratulate the authors for the establishment of the model, but is it worth publishing at this stage? Especially with the results apart from the actual model establishment are somewhat predictable? Apart from the establishment, the results can be summarized with: the bigger spheroids form larger and heavier tumors with more blood vessel connections.
Minor concerns
Line 104: “During the opening of the eggs, they were then examined macroscopically and microscopically for the presence of a tumor by an experienced Sarcoma surgeon.” Please clarify: at the site of the inoculation or were You looking for metastases?
Line 138: “In terms of efferent and afferent vascular connections to vascularize the growing tumor, the largest spheroids were found to present the most vascular connections. Among them, the 100,000 spheroids showed an average of 9.25 […].” It would have been helpful to see a picture of the connections that are counted.
Line 154: “Thus, in the samples of the three groups, all tumor probes taken could be unequivocally identified as Fibrosarcomas.” Do You mean “samples” instead of “probes”?
Line 160: “Figure 6. Immunofluorescence images of Ki67 staining of the same Fibrosarcoma grown from a 100,000 cell spheroid at 4x (left) and 20x (right) magnification.” Is there no way to enhance the exposure?
Line 274: “After cooling down and calculating the required cell solutions, the three cell spheroid sizes were seeded on the agarose-coated 96-well plate to induce spheroid formation.” Please rephrase. If I understand correctly, You have seeded different numbers of cells in suspension in order to get spheroids? “the three cell spheroid sizes were seeded” sounds misleading.
The study is well conducted but seems to report preliminary results, only. I recommend a major revision with additional experiments.
Author Response
First, we would like to thank the reviewer for the valuable suggestions. Please find our point-by-point responses below:
Comment 1:
In my eyes, the main problem of the the study is that it seems to report preliminary results, like a pilot study. HT1080 cells have been used in a CAM model before, just not as spheroids, and spheroids of tumor cells have been used before in a CAM model, just not from fibrosarcomas. Don’t get me wrong, this is important work and I congratulate the authors for the establishment of the model, but is it worth publishing at this stage? Especially with the results apart from the actual model establishment are somewhat predictable? Apart from the establishment, the results can be summarized with: the bigger spheroids form larger and heavier tumors with more blood vessel connections.
Response 1:
We appreciate the reviewer’s critical reflection on the current scope of the study. Indeed, we view this work as a foundational step toward the development of a robust and versatile in ovo model for fibrosarcoma research. While the present manuscript focuses on establishing the feasibility and reproducibility of spheroid-based CAM implantation, we plan further experimental investigations using this model, e.g. pharmacologic testing, angiogenesis modulation, and molecular profiling. Therefore, we consider this publication as a methodological groundwork for a series of subsequent studies that will build upon and extend the utility of the model.
Therefore, our study primarily aimed to establish a reproducible CAM-based model. While CAM models and HT1080 cells have been used separately in previous studies, our work represents – to our knowledge – the first systematic combination of fibrosarcoma-specific spheroid formation and CAM implantation, which could, as already mentioned, serve as a basis for further pharmacologic, vascular, and molecular investigations.
We agree that some findings, such as the correlation between spheroid size and tumor weight or vascularization, may appear expected. Nevertheless, our standardized quantification of vascularization and histological confirmation of fibrosarcoma-specific markers (e.g. Ki67, vimentin) go beyond a mere proof-of-concept, laying the groundwork for a model that may reduce early-phase animal experimentation. We emphasize this more clearly in the revised Introduction and Discussion (see lines 67-70, 199-202, 271-278). Furthermore, we would like to point out that the term “Implementation” in the manuscript title was chosen to clarify the nature of this study as a model-establishing work.
Comment 2:
Line 104: “During the opening of the eggs, they were then examined macroscopically and microscopically for the presence of a tumor by an experienced Sarcoma surgeon.” Please clarify: at the site of the inoculation or were You looking for metastases?
Response 2:
We have clarified that the tumors were assessed macroscopically and microscopically at the site of spheroid placement on the CAM, not systemically for metastases (see line 110).
Comment 3:
Line 138: “In terms of efferent and afferent vascular connections to vascularize the growing tumor, the largest spheroids were found to present the most vascular connections. Among them, the 100,000 spheroids showed an average of 9.25 […].” It would have been helpful to see a picture of the connections that are counted.
Response 3:
We appreciate the suggestion and have added a representative image of the counted vascular connections in the text.
Comment 4:
Line 154: “Thus, in the samples of the three groups, all tumor probes taken could be unequivocally identified as Fibrosarcomas.” Do You mean “samples” instead of “probes”?
Response 4:
Corrected. We now consistently use the term “samples” instead of “probes” throughout the manuscript.
Comment 5:
Line 160: “Figure 6. Immunofluorescence images of Ki67 staining of the same Fibrosarcoma grown from a 100,000 cell spheroid at 4x (left) and 20x (right) magnification.” Is there no way to enhance the exposure?
Response 5:
Thank you very much for this observation. Due to limitations in the original imaging conditions, the exposure could not be further optimized without compromising signal integrity.
Comment 6:
Line 274: “After cooling down and calculating the required cell solutions, the three cell spheroid sizes were seeded on the agarose-coated 96-well plate to induce spheroid formation.” Please rephrase. If I understand correctly, You have seeded different numbers of cells in suspension in order to get spheroids? “the three cell spheroid sizes were seeded” sounds misleading.
Response 6:
We agree that the phrase was unclear and rephrased it for clarity (see lines 302-305).
Reviewer 2 Report
Comments and Suggestions for Authors
This article describes the development of a model system using the CAM assay and fibrosarcoma spheroids for cost-effective, non-animal preclinical research. The study's objective was clear, and the results presented were simple and easy to follow.
Major Points:
- The figures need to be reorganized for better clarity.
- It is recommended to combine Figures 5 and 6.
- Figure 7 appears redundant and could be removed.
- Figure 8 would be better placed in the section where the CAM assay is discussed.
Including a table that details the number of eggs excluded during the study process would be helpful.
Author Response
We thank the reviewer for the constructive feedback and carefully considered each point. We have addressed the concerns in detail below. All changes are highlighted in the revised manuscript.
Comment 1:
It is recommended to combine Figures 5 and 6.
Response 1:
We appreciate the reviewer’s suggestion to merge Figures 5 and 6. After careful consideration, we decided to retain them as separate visual elements, now labeled as Figure 5A and Figure 5B, to preserve clarity and avoid visual overcrowding. In our opinion, the two panels illustrate different aspects of the model and are better appreciated when displayed individually.
Additionally, both subfigures are placed in close proximity within the manuscript, allowing a continuous interpretation.
Comment 2:
Figure 7 appears redundant and could be removed.
Response 2:
We thank the reviewer for the suggestion regarding Figure 7 and removed it.
Comment 3:
Figure 8 would be better placed in the section where the CAM assay is discussed.
Response 3:
We moved former Figure 8 into the Methods section under the CAM assay subsection, where it fits more logically (now Figure 2).
Comment 4:
Including a table that details the number of eggs excluded during the study process would be helpful.
Response 4:
We recognize the value of a tabular breakdown of excluded eggs, we regret to note that due to limitations in retrospective documentation, we were not able to reliably reconstruct the precise distribution of excluded samples across the different spheroid groups. Nevertheless, we have revised the corresponding section in the manuscript to clarify the exclusion process and provide transparent information on the overall number of evaluable eggs. We hope that this revised description improves clarity and adequately addresses the concern. For future experiments, we plan to implement a more detailed exclusion protocol to ensure comprehensive documentation and enable precise reporting in line with reviewer recommendations.
Round 2
Reviewer 1 Report
Comments and Suggestions for Authors
Thank You for the response and clarifications.
Reviewer 2 Report
Comments and Suggestions for Authors
The authors have revised or responded to the points raised in their original article. The revisions are satisfactory, and their response is acceptable. The article appears ready for publication.